# Digital Serious Games for Cancer Education and Behavioural Change: A Scoping Review of Evidence Across Patients, Professionals, and the Public

**DOI:** 10.3390/cancers17203368

**Published:** 2025-10-18

**Authors:** Guangyan Si, Gillian Prue, Stephanie Craig, Tara Anderson, Gary Mitchell

**Affiliations:** School of Nursing & Midwifery, Queen’s University Belfast, Belfast BT7 1NN, UK; g.prue@qub.ac.uk (G.P.); s.craig@qub.ac.uk (S.C.); tanderson@qub.ac.uk (T.A.); gary.mitchell@qub.ac.uk (G.M.)

**Keywords:** serious games, cancer awareness, cancer prevention, health behaviour, health literacy, patient education, healthcare professionals, digital health

## Abstract

Digital serious games are being developed to support cancer education, prevention, care, and survivorship. However, their overall role and impact have not previously been synthesized in a single review. This scoping review examined serious games used across three stakeholder groups: the public, people living with cancer, and healthcare professionals. The Capability, Opportunity, Motivation, and Behaviour model was applied to interpret how these games influence learning and behavioural outcomes. The review found that serious games can increase knowledge, improve engagement, strengthen communication between patients and professionals, and encourage preventive health behaviours. The findings highlight the need for future studies to use rigorous evaluation methods, user-centred design, and implementation strategies to maximize the long-term value of serious games in cancer education and awareness.

## 1. Introduction

According to the World Health Organization (WHO) cancer is the primary or secondary major cause of death before the age of 70 years in 91 of 172 countries, and third or fourth in another 22 countries [1,2]. Globally, the number of new cancer cases is predicted to increase by about 50% over the next 20 years [3]. Cancer prevention and management have become integrated in key health policy globally [4]. The incidence and mortality of cancer are anticipated to escalate significantly due to population growth, ageing, and the adoption of lifestyle habits that raise cancer risk, particularly in countries with low and middle incomes [5]. Poor screening programmes, delays in early detection, and a lack of awareness about cancers contribute to the high incidence and mortality rates [6]. Therefore, raising awareness about cancers, and improving the confidence and self-efficacy among the public may help increase the early detection rate and even lower the mortality rate [6]. Traditional health education typically delivers information through lectures, seminars, written materials, and pamphlets, which have been shown to effectively increase knowledge [7,8]. However, these methods are often criticized for lacking engagement and interactivity [8,9]. The engagement is essential to effective learning and motivation in medical education [10]. Gamification and game-based learning (GBL) have recently emerged as fresh and appealing forms of health education, and they have been shown to perform better in knowledge acquisition than traditional teaching approaches [11].

Serious games are defined as tools designed for purposes beyond mere entertainment, such as education, training, and skill development [12]. Compared to conventional educational methods, serious games appear to offer a more engaging and dynamic approach to learning [13]. They have been shown to enhance health awareness, boost confidence, and support more effective knowledge acquisition [13,14,15]. Serious games have also been utilized in various healthcare fields, such as in medical diagnostics, therapeutic practices, health prevention and promotion, and patient education [16]. A serious game about pancreatic cancer has been shown to be an effective intervention for improving public awareness and increasing self-efficacy of acknowledging cancer symptoms and seeking help intentions [17]. The game, co-designed with experts, patient advocates, and healthcare professionals, presented information about pancreatic cancer through an interactive format. Anderson et al. [17] demonstrated that a digital serious game significantly improved pancreatic cancer symptom awareness and help-seeking intentions among the public. Other studies have also suggested that serious games encourage cancer prevention behaviours by the public—for example, increasing cancer prevention knowledge and motivating self-examination performance—which may contribute to early detection and improved health outcomes [18,19,20]. Digital serious games, in particular, offer the advantage of remote accessibility, scalability, and the potential for real-time feedback and personalisation, making them a versatile tool in public health and clinical education. The potential benefits of serious games are not limited to the public. Cancer patients may benefit from games that enhance self-management and treatment understanding, while healthcare professionals may use them for communication training or continuing education. As such, serious games are increasingly relevant to multiple groups along the cancer care continuum.

This scoping review focused on cancer patients, healthcare professionals, and individuals in the public. One reason for this preference was that the three groups of people may capture a comprehensive perspective on serious games’ performance on improving cancer awareness and education [21]. Cancer awareness involves different stakeholders, including cancer patients, healthcare providers, and public decision-makers [21]. In addition, the gaps in cancer awareness may be identified between the three groups. Cancer patients may lack knowledge about oncological disease and related symptoms, and may experience psychological effects as a result of anxiety about their illness [22]. Healthcare professionals may need continuous education to provide accurate care and treatment and knowledge or improve communication when providing care and treatment [22]. Public members may have myths that delay the early detection of cancer or have poor cancer preventive behaviour [23]. Another reason was this scoping review study aims to map the breadth and depth of existing evidence on cancer awareness and education [24]. Therefore, including various populations helps identify the difference on research progress and serious games used for each group. This review therefore takes a comprehensive, stakeholder-inclusive approach to understanding how serious games are being used across patients, the public, and professionals. By integrating evidence from these distinct but interconnected groups, the review aims to present a broader picture of serious games’ role in cancer education and behavioural support. Focusing specifically on digital serious games enables a clearer understanding of how technology-enhanced interventions can support behaviour change and learning across diverse contexts.

Several relevant literature reviews have been conducted on the use of serious games and digital interventions in health education and cancer management, encompassing both systematic and narrative approaches [21,22,23,24]. Among these, one narrative review, one rapid review, and one integrative review have been published [25,26,27], alongside two systematic reviews [28,29]. These reviews have addressed a range of topics including methods to improve treatment adherence [21], digital tools for skin self-examination [22], digital games for cancer management [25], and serious games aimed at improving knowledge and self-management among young people with chronic diseases [24]. Other reviews focused on games supporting oncological disease self-management [26], pain management in cancer patients [27], and digital interventions targeting adolescents and young adults with cancer [23]. The populations studied in these reviews vary widely, including adolescents and young adult cancer patients [21,22,23], young people with chronic diseases [24], cancer patients across all age groups [27], and the public [22]. The systematic reviews have explored gamification as an educational tool for cancer patients [28,29], digital interventions for cancer prevention in people with disabilities [28], and gamification for self-management in chronic disease populations [29]. While most existing reviews focus on digital interventions for cancer patients, fewer have examined serious games aimed at cancer prevention and awareness in the broader public. To date, no review has integrated the perspectives of patients, public, and healthcare professionals within a single framework, creating a unique opportunity to compare outcomes across stakeholder groups. Importantly, no scoping review to date has comprehensively mapped the empirical research on digital serious games designed specifically to promote cancer awareness and understanding across multiple population groups, including patients, healthcare professionals, and the public.

To address this gap, the current scoping review aims to systematically map the existing evidence on digital serious games used for cancer awareness, education, and behavioural support among adults. This review focuses on three distinct stakeholder groups: cancer patients, healthcare professionals, and the public. The purpose is to capture a comprehensive understanding of how serious games are being utilized across different stages and contexts of the cancer care continuum. The objectives of this review are to:Examine the current evidence on digital serious game interventions aimed at improving cancer-related education, awareness, and behavioural outcomes among adults.Explore the reported outcomes of digital serious games across adult cancer patients, healthcare professionals, and the public, including impacts on knowledge, engagement, confidence, communication, self-efficacy, and behavioural intentions.Identify gaps in the literature concerning the design, delivery, and evaluation of digital serious games for cancer education and awareness across diverse stakeholder groups.

Highlights:Global cancer burden: Cancer is a leading cause of death worldwide, with incidence and mortality expected to rise due to population growth, ageing, and lifestyle factors.Importance of awareness and education: Improving public, patient, and healthcare professional awareness and self-efficacy may contribute to early detection, better prevention behaviour, and reduced mortality.Potential of digital serious games: Compared to traditional education methods, digital serious games provide more interactive and engaging approaches to improve cancer education, knowledge acquisition, cancer awareness, self-efficacy, and help-seeking intentions.Stakeholder diversity: Digital serious games can benefit multiple groups, including cancer patients (self-management), healthcare professionals (training, communication), and the public (awareness and prevention).Evidence gap: While previous reviews have examined serious games in specific contexts, no review has comprehensively integrated perspectives from patients, healthcare professionals, and the public on digital serious games for cancer education and behavioural change.Scope of current review: This scoping review aims to systematically map existing evidence on digital serious games for cancer awareness, education, and behavioural change among adults, highlighting gaps and informing future research and design strategies.

## 2. Materials and Methods

### 2.1. Protocol and Registration

This scoping review was conducted following the Joanna Briggs Institute (JBI) methodology for scoping reviews [25] and is reported in accordance with the PRISMA-ScR (Preferred Reporting Items for Systematic Reviews and Meta-Analyses extension for Scoping Reviews) checklist [30]. The completed PRISMA-ScR checklist is available in Appendix A. A protocol for this scoping review was registered in a prospective manner on Open Science Framework on 30 April 2025 (registration DOI: https://doi.org/10.17605/OSF.IO/YM6TX).

### 2.2. Eligibility Criteria

Both experimental and quasi-experimental study types, such as randomized controlled trials, non-randomized controlled trials, before and after studies, and interrupted time-series studies, were considered for inclusion in this scoping review. Additionally, case–control studies, analytical cross-sectional studies, prospective and retrospective cohort studies, and other analytical observational research were also considered for inclusion. Qualitative studies that primarily report qualitative data were also taken into consideration. These studies included, but were not restricted to, action research, grounded theory, ethnography, phenomenology, and qualitative description. Relevant studies cited in the reference lists of identified systematic reviews were also screened for inclusion. Studies published in English only were included as there is a lack of language diversity amongst the researchers. There were no publication date restrictions for the inclusion, and grey literature was excluded to maintain the rigour and reliability of the findings; however, this may have limited the comprehensiveness of the evidence base. Additional eligibility criteria were formulated in accordance with the Population, Concept, and Context (PCC) framework, as recommended by the JBI [25].

#### 2.2.1. Population

This scoping review focused on adult participants from the public, including cancer patients, healthcare professionals, students, and individuals from diverse occupational backgrounds, without restrictions on gender, ethnicity, and education level. Studies involving individuals under 18 years were excluded. The reasons for this exclusion are the experiences, developmental stages, healthcare needs, and legal considerations of children and adolescents differ significantly from those of adults and often requiring separate frameworks and interventions [26]. However, studies including mixed-age samples (e.g., adolescents and adults) were considered eligible only if adults constituted more than 50% of the participants and if data for the adult subgroup could be clearly extracted and analyzed separately [27]. This approach ensured consistency with the Population component of the PCC framework while allowing inclusion of studies where adult data were meaningfully represented.

#### 2.2.2. Concept

This review focused on studies examining the impact of digital serious games on cancer awareness, education, and related outcomes. Digital serious games are defined as purpose-built, interactive games delivered through electronic platforms (e.g., mobile apps, computer programmes, web-based tools) that aim to inform, educate, or influence behaviour beyond entertainment. For this review, cancer awareness was defined broadly to include knowledge or beliefs about cancer symptoms, risk factors, early detection, treatment effectiveness, and prevention strategies [28,29]. Studies were included if they explored how digital serious games enhanced understanding of cancer-related topics or supported educational outcomes. Research involving any type or stage of cancer was considered eligible. Both qualitative and quantitative studies were included, provided they examined experiences or outcomes related to cancer awareness and education. Studies could involve any population group including patients, healthcare professionals, caregivers, students, or members of the public, reflecting the multi-stakeholder approach of this review. Interventions that used digital serious games to improve knowledge, support behaviour change, encourage help-seeking, or enhance communication about cancer were all within scope.

#### 2.2.3. Context

This review included serious game interventions delivered through mobile, computer, and digital platforms. Serious games are defined as tools designed for purposes beyond mere entertainment, such as education, training, and skill development [12]. Digital delivery enables scalable, flexible, and interactive formats, which are particularly important in health education and behaviour change interventions. Studies focusing on digital serious games aimed at promoting cancer awareness and education were eligible for inclusion. In contrast, articles examining other cancer awareness interventions, such as leaflets, pamphlets, or non-digital serious games were excluded. Research settings varied and could include healthcare environments or any public spaces. Studies from all geographical locations were considered.

### 2.3. Search Strategy

An initial limited search of MEDLINE and PsycINFO, Web of Science, and CINAHL was undertaken to identify articles on the topic. The text words contained in the titles and abstracts of relevant articles, and the index terms used to describe the articles were used to develop a full search strategy for the four databases (Appendix A). The search strategy, including all identified keywords and index terms, was adapted for each included database and/or information source. A systematic search was conducted on 10 August 2025, across the following databases to identify potentially relevant studies: MEDLINE, PsycINFO, Web of Science, and CINAHL. The reference list and citations of all included sources of evidence were screened for additional studies. To ensure the inclusion of the up-to-date evidence, an additional search was performed before manuscript submission. This updated search did not identify any new eligible studies beyond those captured in the original search.

### 2.4. Selection of Sources of Evidence

Following the database search, all identified citations were imported into Covidence (https://app.covidence.org, accessed on 30 April 2025), a web-based tool used to streamline the screening and data extraction process. Duplicate records were automatically detected and removed during the importing process. Title and abstract screening were conducted independently by two reviewers (GS and GM) using the predefined inclusion criteria. Full-text articles were then assessed for eligibility independently by the same two reviewers. Reasons for exclusion at the full-text stage were documented and are reported in this review. Any disagreements at any stage of the screening process were resolved through discussion among other members of the review team (GP, TA, and SC).

### 2.5. Data Charting

Covidence was used to chart the data. The ‘JBI template source for evidence details, characteristics and results extraction instrument’ [25,30] was followed and modified for this scoping review. The current scoping review extraction form also summarized study design, study objectives, and types of serious games. GS extracted the data from studies included in the scoping review, and it was checked for accuracy by GM independently. The data extracted included specific details about the author, year, setting, study design, study objectives, study methods, population characteristics, game types, and key findings. In accordance with JBI guidance for scoping reviews, a formal quality appraisal of included studies was not undertaken, as the purpose of a scoping review is to map the breadth of available evidence rather than assess the methodological quality of individual studies [25,30]. Several articles were identified as reporting on the same datasets. When multiple articles described the same underlying study, data were counted as a single study to avoid duplication. All relevant information from companion papers (e.g., different outcomes or populations) was extracted and synthesized together.

### 2.6. Data Analysis

Data were analyzed using a narrative synthesis approach, which is particularly appropriate for scoping reviews that incorporate a wide range of study designs, methodological approaches, and outcome measures [25,29,30]. This analytical method allowed for the integration of both qualitative and quantitative evidence, facilitating the identification of shared patterns across the included studies examining digital serious games for cancer awareness and education. Narrative synthesis was selected for its capacity to accommodate methodological diversity and support the interpretation of findings across varied populations, including cancer patients, healthcare professionals, and members of the public. To maintain transparency and analytical rigour, the review process was comprehensively documented. The documentation included maintaining reflective notes to mitigate potential bias and ensure clarity in the progression of analytic decisions. The synthesis was conducted in three sequential phases. In the first phase, data were systematically extracted using a standardized template to ensure consistency in reporting study characteristics and outcomes. The second phase involved thematic exploration to identify recurring concepts related to knowledge acquisition, behavioural change, user engagement, and implementation challenges. In the third phase, the findings were refined and categorized according to the three primary stakeholder groups. This structure was designed to support a clear and coherent presentation of the results and to align with the review objectives.

Collectively, these methods ensured that the evidence was comprehensively mapped and that analytical transparency was maintained throughout the review process. The approach provided a structured foundation for identifying key patterns across study designs and populations, ensuring that the synthesis accurately reflected the scope of current research on digital serious games for cancer education and behavioural change.

## 3. Results

A total of 6596 results were retrieved from the initial search strategy. After removing the duplicates, 5224 studies remained for screening. Following the title and abstract screening, 5168 studies were not relevant to this scoping review, since they did not meet the inclusion criteria. Reasons for exclusion during the screening process included studies which explored different research questions, different populations (e.g., children and adolescents), and different outcomes (e.g., impact on practicing physical activities); and studies which were not in English language and not available in full text (Appendix B). Full-text screening resulted in 31 eligible articles, and an additional four were identified through reference list screening, bringing the final total to 35 articles included in the review. Note: Three studies from the USA, Canada, and Australia reported on the same young cancer patient dataset [31,32,33], and two UK studies shared data from a serious game for African Caribbean men [34,35]. To avoid double-counting, these were treated as single studies during synthesis, with data from both sources combined for completeness. Therefore, 35 articles reporting on 33 unique studies were included. A PRISMA-ScR flow diagram (Figure 1) presents the full study identification process.

### 3.1. Characteristics of Included Studies

The 35 included studies were published between 2006 and 2025 (Appendix A), with most published in the last ten years (*n* = 26). Since 2018, at least two papers have been published each year, with the highest output observed in 2018, 2019, 2023, and 2024, each yielding four publications. Among all the included studies, three were qualitative research [36,37,38], 18 were quantitative research [7,18,19,31,32,33,39,40,41,42,43,44,45,46,47,48,49,50], four were mixed methods studies [20,51,52,53], three used co-design method [17,34,35], and seven were design-based research [54,55,56,57,58,59,60]. Design-based research is an approach created for educators with the goal of enhancing the influence, transmission, and conversion of educational research into effective practice [61]. It emphasizes the necessity of developing theories and design principles that guide, inspire, and enhance research and practice in the field of education [61]. The included studies were carried out in various countries. A majority of the publications were produced in the USA (*n* = 16) [18,38,39,41,42,44,46,47,48,50,51,52,53,56,58,60], and studies in the UK ranked second (*n* = 5) [17,34,35,45,55]. Participants in three studies came from the USA, Canada, and Australia [31,32,33]. The following nations were also included in the included studies: Portugal (*n* = 2) [19,57], Brazil (*n* = 2) [37,59], Norway (*n* = 1) [20], Lebanon (*n* = 1) [54], Australia (*n* = 1) [36], Malaysia (*n* = 1) [43], Thailand (*n* = 1) [7], Taiwan, province of China (*n* = 1) [40], Republic of Korea (*n* = 1) [49].

Participants included members of the public (such as university students and caregivers), healthcare professionals (such as nurses, physicians, and researchers), and cancer patients. Most of the participants in the included studies were members of the public (*n* = 17) [7,17,18,19,20,34,35,36,41,42,43,44,45,46,48,51,56] and followed by cancer patients (*n* = 10) [31,32,33,39,47,49,50,52,53,55]. Six studies investigated healthcare professionals [37,38,58,59,60] and medical students [54]. Cancer patients and healthcare professionals were both explored in three studies [40,56,57]. The included studies’ sample sizes varied from three [37] to 1205 participants [18]. Four design-based research studies did not report the exact number of participants, but the participants characteristics were clarified, they were healthcare professionals [59,60] and medical students [54], and African and African Caribbean men diagnosed with prostate cancer [55].

In addition, the included studies covered a wide range of cancer types, which contained prostate cancer (*n* = 4) [34,35,53,55], melanoma (*n* = 4) [18,19,44,48], cervical cancer (*n* = 4) [20,42,46,51], skin cancer (*n* = 3) [7,36,59], breast cancer (*n* = 2) [43,49], lung cancer (*n* = 2) [38,54], oral cancer (*n* = 1) [40], thyroid cancer (*n* = 1) [37], pancreatic cancer (*n* = 1) [17]. Nine studies examined various types of cancer: four had acute leukemia, lymphoma, and soft-tissue sarcoma [31,32,33,52]; two had breast, cervical, and colon cancer [58,60]; one had lung, breast, and uterine cancer [39]; one had metastatic breast cancer and advanced gynecologic cancer [47]; one had lung and gastrointestinal cancer [50]. However, the specific type of cancer was not indicated in five of the articles [41,45,56,57].

Furthermore, a variety of serious game types were explored by the included studies. Most of the games (*n* = 21) were online serious games which could be accessed with computer, tablet, and mobile phone [7,17,18,20,34,35,39,43,44,45,46,47,48,49,50,51,56,57,59,60], but some serious games were presented in different format and style, such as digital Tamagotchi-style game (*n* = 1) [37], interactive style game (*n* = 3) [38,42,53], video game (*n* = 5) [31,32,33,41,52], simulation-based game (*n* = 2) [54,58], and intelligent game (*n* = 1) [55]. Virtual reality (VR) game (*n* = 2) [36,40] and augmented reality (AR) game (*n* = 1) [19] were also reported.

For the qualitative studies, focus group interview and semi-structured interview were carried out in two studies to participants [36,38]. Oliveira et al. [37] used a qualitative assessment form to collect participants’ opinions on the game’s usefulness, educational value, medical accurateness, and effectiveness. For the quantitative studies, pre- and post- tests were frequently utilized to evaluate the effectiveness of the serious game [7,18,40,43,45,48]. Randomized controlled trials were also commonly used to assess the performance by comparing outcomes with and without playing serious games [31,32,33,39,41,42,44,49,50]. Three studies used post-game questionnaires to explore the feasibility of a game [19,46,47].

The results of this scoping review are organised thematically and presented according to three distinct population groups: cancer patients, healthcare professionals, and members of the public. This structure was chosen to reflect the differing roles, informational needs, and experiential contexts of each group in relation to digital serious games for cancer awareness and education. Presenting the findings in this disaggregated format facilitates a more nuanced understanding of how these interventions are developed, implemented, and perceived across varied populations and healthcare settings. This approach is consistent with the guidance provided by the Preferred Reporting Items for Systematic Reviews and Meta-Analyses extension for Scoping Reviews (PRISMA-ScR), which supports structuring results in a way that aligns with the review objectives and enables meaningful synthesis across heterogeneous evidence [30]. A final integrated discussion follows to bring together the thematic findings and examine how they contribute to the existing research landscape.

### 3.2. The Role of Serious Games in Cancer Awareness Among Patients

Findings from the included studies were structured around three key thematic areas: (1) effectiveness for health and behavioural outcomes, (2) effectiveness in improving cancer knowledge and raising awareness, and (3) challenges related to engagement and personalisation. These themes reflect the breadth of reported outcomes and experiences across the patient population included in this review.

#### 3.2.1. Effective for Health and Behavioural Outcomes

Serious games were associated with improvements in health and behavioural outcomes for cancer patients. Through engagement with serious games, patients can increase their awareness of cancer and support improvements in health behaviours. Evidence for these benefits was reported in seven out of ten studies involving cancer patients. Six were quantitative studies [31,33,39,47,49,50], and one was a mixed-method study [53].

Serious games supported cancer patients in managing treatment-related side effects, such as chemotherapy-induced nausea and vomiting. This process helped cancer patients improve their awareness by taking actions. Such behavioural changes enabled patients to develop decision-making skills and increase self-efficacy related to cancer knowledge. Loerzel et al. [39] demonstrated that older cancer patients aged 60 to 84 years old found serious gaming to be both beneficial and acceptable. The participants who played the game drank more fluids at all time points compared to those who did not play the game. In comparison to the control group, the game group reported applying almost twice the number of chemotherapy-induced nausea and vomiting (CINV) preventive measures (1486 versus 768). However, the control group reported using nearly twice as many CINV self-management measures (1311 versus 681) as the game group. A more in-depth explanation revealed that a higher percentage of participants in the game group engaged in reporting or monitoring the preventive measures that they took at home throughout the research process. They mentioned utilizing greater food strategies, medications, and relaxation/distraction approaches to prevent CINV compared to the control group. Therefore, participants in the game group may have been able to avoid CINV by using preventative measures, which reduced the need for self-management techniques to treat CINV symptoms. Additionally, video-based serious games for self-care interventions would be a helpful psycho-educational tool for improving treatment outcomes for cancer patients [33]. Beale et al. found the mean acceptability rating was 4.1 on a 5-point scale, indicating a high level of acceptability; the mean credibility rating was 3.7, suggesting a moderate level of perceived effectiveness of the game as a psycho-educational intervention [33]. Correlational analyses also revealed that both acceptability and credibility ratings were significantly and positively associated with the amount of time spent playing the video game during treatment (acceptability: *r* = 0.26, credibility: *r* = 0.25, *p* < 0.01).

Serious games also contribute to improved treatment adherence, quality of life, medication compliance, and self-efficacy. Serious games not only improved behavioural outcomes but also served as effective tools for raising cancer awareness, as they engaged cancer patients through interactive content that reinforces key messages and supports informed health decision-making. Serious games’ engagement and satisfaction were generally high among cancer patients [47,49,50,53]. Kato et al. reported an increase of 16% in treatment adherence for the game group [31]. Better drug adherence was demonstrated by the mobile gaming group [49]. Compared to the control group, the mobile game’s users experienced fewer adverse effects from chemotherapy, including fatigue, nausea, numbness in hands or feet, and hair loss. During treatment, the game group showed improved quality of life (QoL). Analysis using mixed-effect linear models also showed that participants in the game and control groups had comparable baseline levels of cancer-specific self-efficacy, but that the game group’s self-efficacy increased over time to a significantly greater extent [31]. In addition, compared to participants who were comparatively less engaged, highly engaged participants reported noticeably greater 3-month self-advocacy abilities of connected strength [47]. According to You et al., participants who engaged in fewer repetitions of game scenarios reported higher symptom severity at baseline. Playing games also helped patients manage their real-life desires to smoke [50]. Serious games have been shown to enhance patients’ decision-making by encouraging them to ask more insightful and informed questions during consultations with their physicians, while also providing updated information on potential treatment side effects [53]. The absence of reported adverse effects suggests that the serious game is safe for patient use [31].

#### 3.2.2. Effective in Educating Cancer Knowledge and Raising Awareness

Serious games were found to enhance cancer-related knowledge and symptom awareness among patients. Support for this finding was found in nine studies, comprising one qualitative study [37], four quantitative studies [31,32,33,40], one mixed-methods study [52], and three design-based studies [55,56,57].

A randomized controlled trial (RCT) of a video game that improves behavioural outcomes in adolescents and young adults with cancer suggested although both game groups’ or control groups’ initial levels of cancer-related knowledge were similar, according to mixed-effect linear model analyses, the game group showed significantly greater gains in knowledge over time [31,32,33]. Findings suggested participants’ perceptions of the game’s acceptability and credibility were not significantly associated with their knowledge gains following gameplay, which was further explained as positive evaluations of the game as an enjoyable activity or therapeutic tool were independent of its demonstrated effectiveness as a knowledge acquisition instrument [32]. A test was conducted to determine whether playing a serious video game was considered a preferred way to learn about cancer. Results showed that playing a serious game was preferred by the largest percentage of patients (23%), compared to searching the internet (21%), consulting a physician (19%) or asking another patient (19%) [52]. Another RCT also reported that patients in a VR game group were more likely to express that the VR-based game met their needs, conveyed knowledge clearly, and that they would be willing to use them again (93.8%), compared to patients in a written format group (57.1%) [40]. In addition, overall satisfaction was higher in the VR group (63%) than in the written aids group (54%) [40]. A Tamagotchi-style serious game designed for thyroid cancer patients simulated the care of a virtual thyroid pet with related health conditions. The game was positively evaluated by thyroid health specialists, who praised its educational value, accurate depiction of thyroid diseases, and high-quality visual design [37]. Expert assessments highlighted the game’s comprehensive clinical scenarios and realistic animations, which effectively conveyed medical information through an intuitive interface and visually engaging graphics. However, its effectiveness in improving cancer awareness among patients has not yet been tested.

Design-based research following a structured methodology explored the process of creating a digital serious game, designing game content, and evaluation of the game. After evaluation among African and African-Caribbean men with prostate cancer, an intelligent serious game was designed to provide knowledge about prostate cancer and was able to encourage prostate cancer patients who experienced cancer at all stages to seek medical assistance [55]. In addition, through the utilization of a formative evaluation process and focus groups, a serious game was reacted positively by participants who included cancer patients and nurses for its effectiveness as an educational intervention for patients [56]. The results of a usability testing study of a mobile app for health literacy and self-management among oncological patients showed that another serious game also received favourable feedback after being evaluated by 132 cancer patients and healthcare professionals [57]. Gamified features such as reward systems and quizzes were particularly valued for their effectiveness in continuous user motivation and engagement [57].

#### 3.2.3. Challenges in Engagement and Personalisation

Although serious games were effective in acquiring knowledge and raising cancer awareness, some challenges were identified in relation to engagement and personalisation. Support for this theme was found in five studies, comprising one qualitative study [38], three quantitative studies [33,49,50], and one mixed-methods study [53].

Some participants expressed concern that the emotional content of the serious game might be overly distressing when experienced immediately prior to an appointment at a cancer clinic [38]. Patients with lung cancer have reported feeling “shellshocked” during their initial visits to the oncology office [38]. No significant differences in anxiety or depression scores were observed among 76 breast cancer patients [49]. A study which investigated young cancer patients’ perceptions of a video game used to promote self-care stated the completion rate of the game was low, possibly due to the difficulty of the gameplay, particularly considering that many patients were coping with serious illness and experiencing side effects of treatment [33].

The timing of implementing serious games as adjunctive therapy for cancer patients is critical; for instance, during hospitalization, it may be inappropriate to introduce games aimed at smoking cessation [50]. This may be attributed to the framing of the application as a ‘game,’ which could have diminished user receptivity, particularly given the potential distraction caused by hospitalization requirements and health-related concerns [50]. Negative findings were primarily related to the implementation of specific game features. In a study assessing the acceptability and usability of an interactive serious game in aiding treatment decisions for patients with prostate cancer, the generally positive evaluations of the game’s overall concept, contrasted with less favourable responses regarding its personal relevance, suggest a broader need for increased personalization [53].

### 3.3. The Role of Serious Games in Cancer Awareness Among Healthcare Professionals

Compared to studies focused on patients, evidence on serious games targeting healthcare professionals was more limited and diverse in nature. A total of six studies specifically involved healthcare professionals and students [37,38,54,58,59,60], with an additional three studies featuring mixed samples that included healthcare professionals alongside patients or members of the public [40,56,57]. While the scope and design of these studies varied considerably, a dominant focus across this body of work was the educational value of serious games, particularly in supporting cancer-related knowledge acquisition, training, and confidence among healthcare providers. Though a smaller subset of the literature also touched on communication, the strength and consistency of findings in this area were more limited. Consequently, the findings in this section are synthesized under one core theme related to educational and professional benefits, with communication considered a related, but less commonly evaluated, outcome.

Across multiple studies, serious games were shown to enhance knowledge, skills, and confidence among healthcare professionals in cancer care contexts. These outcomes were reported in studies using a range of methodological designs, including one quantitative study [40], one mixed-methods study [52], and five design-based studies [54,57,58,59,60]. This focus on design-based methodologies reflects the iterative and exploratory nature of developing and evaluating digital training tools for clinical application.

In one mixed-sample study, a virtual reality (VR) serious game was introduced to nurses working in oncology settings [40]. Participants reported substantial gains across several professional domains. Specifically, self-rated confidence in providing cancer-related educational support rose from a baseline of 85 to 98, familiarity with new digital educational tools increased from 36 to 95, and knowledge regarding treatment decision-making improved from 28.5 to 100. Most participants (88–95%) indicated that they agreed or strongly agreed that the VR tool enhanced the quality of bedside pre-treatment education and expressed willingness to recommend it to colleagues. These findings suggest that immersive, digital environments can provide highly acceptable and effective training formats for clinical education, especially when time and access to conventional training may be limited. Similarly, a mixed-methods study evaluating a serious game with healthcare professionals found the intervention improved knowledge about cancer symptom recognition and enhanced preparedness to educate patients [52]. Participants valued the ability to engage with evidence-based content in a format that encouraged active learning and reflection, supporting prior research highlighting the importance of interactive learning modalities in healthcare professional development.

The educational promise of serious games was further supported by a group of studies using design-based approaches. A simulation-based game developed for medical students and professionals was described as offering critical value in promoting diagnostic accuracy and reducing misinformation during patient consultations [54]. This was achieved by embedding real-life clinical decision-making tasks and scenarios into gameplay, allowing learners to rehearse complex interactions in a safe, repeatable environment. These simulation tools were not only engaging but also aligned with competency-based medical education principles, providing a mechanism for skills development that mirrors real-world demands. Another study developed and tested a serious game aimed at improving health literacy among oncology patients, which also included healthcare professionals as co-users in its evaluation [57]. Participants rated the tool positively for its usability, relevance, and potential to facilitate shared understanding between clinicians and patients. Key features such as gamified quizzes and personalised content were identified as helpful in sustaining user engagement, while also reinforcing critical knowledge areas. The inclusion of healthcare professionals in the game’s development and testing phases helped ensure that clinical accuracy and practical utility were embedded in the tool’s design.

Two additional studies focused on the use of simulation games for cancer screening education among healthcare professionals [58,59]. These studies found that online serious games were feasible to use independently and flexibly within clinical workflows. The games were praised for their ability to support information retention, increase motivation, and help users apply theoretical knowledge to practical screening decisions. Furthermore, a subsequent study [60] reinforced these findings by demonstrating strong user satisfaction and perceived training value among healthcare staff engaged in cancer screening education. Collectively, these studies demonstrate how digital games can serve as a scalable and flexible tool for continuing professional development, particularly in fields such as oncology where staying current with guidelines and innovations is essential.

While education was the dominant outcome across this body of evidence, a smaller number of studies suggested that serious games may also support communication development among healthcare professionals. One qualitative study [38] specifically explored how a serious game could facilitate the practice of complex patient-provider dialogues. Participants noted that the game provided opportunities to simulate challenging conversations, such as breaking bad news or discussing treatment risks. These interactions were reported to enhance confidence in real-world communication and decision-making. However, the development of communication was not extensively addressed in the wider literature, and few studies directly measured communication outcomes. Where mentioned, communication-related outcomes were often embedded within broader evaluations of user satisfaction or general training impact, rather than assessed through targeted outcome measures. Given this limited scope, communication is best understood as an emergent benefit rather than a consistently evaluated outcome in the current literature. Nonetheless, its inclusion highlights the potential for serious games not only to build knowledge and confidence, but also to create a reflective space where interpersonal and relational skills can be practised and refined.

### 3.4. The Role of Serious Games in Cancer Awareness Among the Public

Compared to patients and healthcare professionals, members of the public engage with serious games from a different perspective, typically as non-specialists who may lack prior knowledge of cancer symptoms, risk factors, or screening procedures. For this group, cancer awareness involves not only acquiring factual information but also overcoming emotional and motivational barriers to engagement, such as fear, stigma, or perceived invulnerability. Accordingly, the included studies focused on evaluating serious games that target education, prevention behaviours, and health-seeking intentions in public users. The findings across 16 studies involving public participants were synthesised into two key thematic areas: improving awareness, education, and behavioural change, and considerations on user experience and game design.

#### 3.4.1. Improve Awareness, Education, and Behavioural Change

The public participants across the reviewed studies consistently reported high levels of satisfaction and engagement with serious games. These interventions were frequently described as effective in enhancing cancer knowledge, increasing symptom recognition, promoting self-efficacy, and motivating positive health behaviours such as help-seeking and self-examination. This theme was supported by one qualitative study [36], ten quantitative studies [7,18,19,41,42,43,44,45,46,48], two mixed-methods studies [20,51], and three co-design studies [17,34,35].

Numerous studies highlighted that serious games were perceived as more engaging than traditional education formats such as pamphlets or written materials. Participants consistently rated their experience with serious games as more enjoyable, interactive, and motivating [7,17,19,20,35,36,42,46,51]. In comparative studies, those who used serious games reported higher enjoyment scores than those exposed to pamphlets [7,44,46,48]. For example, in one study comparing educational formats for skin cancer, the game group demonstrated higher levels of enjoyment and engagement than the pamphlet group, even though knowledge accuracy was similar [48]. This suggests that games may not only facilitate knowledge retention but also create more favourable learning environments that encourage sustained attention and behaviour change.

Several studies focused specifically on self-screening and symptom recognition. Serious games were used to teach and motivate self-examination for conditions such as melanoma, cutaneous malignancies, and cervical cancer [7,18,19,44,51]. These games often incorporated visual demonstrations of screening procedures, interactive decision-making tasks, or storytelling elements to reduce anxiety and enhance self-efficacy. In a cervical cancer game, players who experienced an in-game demonstration of a screening procedure reported lower anxiety and increased willingness to undergo real-world testing [51]. However, user feedback also revealed that poorly tailored feedback mechanisms within games, such as receiving incorrect or unexpected results, could undermine confidence in identifying symptoms, highlighting the need for careful instructional design [18].

A number of co-designed and mixed-methods studies highlighted improvements in help-seeking intentions following gameplay [17,34,35,41]. In one study focused on pancreatic cancer, members of the public who played a serious game co-designed with clinicians and patients demonstrated greater willingness to consult a doctor when experiencing potential cancer symptoms [17]. Other studies targeting prostate cancer similarly reported that public users felt more motivated to seek medical help and more confident in identifying when to do so [34,35]. These findings suggest that serious games may have value not only in improving factual understanding but also in shifting attitudes and intentions, which are essential precursors to preventive health action. Demographic differences were also observed in gameplay preferences and knowledge gains. For instance, younger participants and women were more likely to engage with serious games on cervical cancer and reported higher usability scores [42,46]. However, older adults and immigrant women were sometimes reluctant to play or needed support to navigate the digital interface, indicating the need for more inclusive and accessible game designs [46]. Despite these limitations, post-game assessments showed significant improvements in cancer knowledge among most users, with many studies reporting an increase in correct responses in knowledge tests after gameplay [45]. Overall, serious games were found to effectively support cancer education in public populations by translating complex health information into accessible, engaging, and emotionally resonant formats. Their flexibility, digital availability, and potential for customisation also make them well-suited for large-scale health promotion efforts [43].

#### 3.4.2. Considerations on User Experience and Game Design

While serious games were largely well-received by the public, several design and usability issues were identified that may influence their broader uptake and effectiveness. These included physical discomfort, technological barriers, cultural differences, and preferences in content delivery. These considerations were documented in one qualitative study [36] and one quantitative study [46].

One common issue related to physical discomfort, especially in games using virtual reality (VR). In a study evaluating a VR game on cancer education, 11% of users reported nausea during gameplay [36]. This side effect could limit the accessibility of immersive technologies for some users, particularly older adults or those with sensory sensitivities. Additionally, hygiene concerns were raised about the shared use of physical game controllers, particularly in public or clinical settings [36]. These practical barriers highlight the importance of considering infection control and accessibility in game deployment strategies.

Cultural and demographic factors also affected engagement. A study comparing public responses across different ethnic groups found that Korean participants were less engaged with a cervical cancer game compared to English and Vietnamese users [46]. The Korean group reported lower scores on the entertainment dimension of the game and spent significantly less time playing. This was mirrored in the perception of educational effectiveness, where 91% of English and 97% of Vietnamese participants considered the game useful, compared to 68% of Korean users. These findings suggest that cultural preferences and gaming familiarity may shape the perceived value of serious games, reinforcing the need for tailored content development.

User feedback also provided suggestions for improving game design. Participants recommended extending gameplay experiences with visual transitions, more diverse storylines, and explicit connections between lifestyle choices and cancer risks [36]. This feedback underscores the role of user-centred design in enhancing both the impact and appeal of serious games for public health education.

In conclusion, while the public demonstrated high levels of satisfaction and educational benefit from digital serious games, usability, cultural appropriateness, and physical design considerations are essential to maximize their reach and effectiveness. By addressing these key factors-usability, cultural appropriateness, and physical design, developers can ensure that serious games serve as inclusive and impactful tools for cancer awareness in diverse community settings. Taken together, the results demonstrate that digital serious games have been implemented across a wide range of cancer types, populations, and platforms, with generally positive outcomes for knowledge, engagement, and self-efficacy. Evidence indicates that these interventions can enhance patient understanding and adherence, strengthen professional education and communication, and improve public awareness and preventive intentions. However, the overall evidence base remains heterogeneous, and few studies have examined sustained behavioural change or long-term effects. The findings therefore suggest considerable potential for digital serious games as an emerging strategy for cancer education, while highlighting the need for more rigorous evaluation and theoretical integration.

## 4. Discussion

This scoping review identified and synthesized evidence on the use of digital serious games for cancer awareness and education among three populations: patients, healthcare professionals, and the public. The findings demonstrate that serious games may influence a wide range of behavioural, cognitive, and emotional outcomes across these groups. When interpreted through the COM-B model (Figure 2) [62,63,64,65], which conceptualizes behaviour as an interaction between Capability, Opportunity, and Motivation, the evidence reveals critical levels and gaps in digital game interventions for cancer-related behaviours. The COM-B framework is particularly well-suited for this synthesis given its empirical grounding and broad application across contexts [66,67]. During data extraction, each included study was mapped to the COM-B model based on the primary behavioural mechanisms targeted by the serious game intervention. The COM-B components were indicated in the data characteristics table (Appendix A).

### 4.1. Capability

Psychological capability refers to the individual’s knowledge and psychological skills to perform a behaviour. Across all three target populations, digital serious games demonstrated strong potential to enhance cancer-related knowledge, thereby increasing psychological capability. Cancer patients showed improved understanding of treatment side effects, symptom management, and health self-advocacy [33,39]. Similarly, public-facing games enhanced symptom recognition, screening literacy, and intentions to seek help [41,48]. Among healthcare professionals, educational simulations led to increased confidence in delivering patient education and recognizing early signs of cancer [40,58]. These findings align with prior evidence on the importance of tailoring capability-enhancing content to the behavioural target [68]. Moreover, educational improvements were not uniform. In public-facing interventions, differential gains were observed by ethnicity and age group, with Korean participants reporting lower acceptability and perceived learning compared to English and Vietnamese participants [46], suggesting variability in how psychological capability is influenced by cultural or demographic factors. Physical capability was less frequently addressed, which is not unexpected given the cognitive nature of most cancer awareness behaviours. However, when games simulated physical tasks, such as self-examinations for skin or breast cancer, there was some evidence of increased confidence in performing these behaviours [7,33], though this area remains under-researched.

### 4.2. Opportunity

Opportunity in the COM-B model refers to all the external factors that make a behaviour possible or prompt it. Physical opportunity includes access to resources, time, and technology [63]. In this review, games were often delivered through mobile apps, VR, or web-based platforms, making them relatively accessible and scalable. However, issues such as motion sickness from VR [36] and concerns around hygiene in shared settings were raised, suggesting that while technological delivery can create opportunity, it can also constrain it when not carefully implemented. Social opportunity includes interpersonal influences, cultural norms, and social support. Few studies in the review explicitly designed interventions to harness social opportunity, although some games involved role-play or decision-making scenarios that mimicked real-world interactions. For example, simulations for healthcare professionals created scenarios for communication with patients, and some public games encouraged discussion around sensitive topics such as cervical screening [7,40]. According to Damschroder et al. [69], the incorporation of social opportunity elements into intervention design can increase sustainability and adoption of behavioural practices. However, this was not a major feature of most interventions in this review and represents a critical implementation gap. Furthermore, Wang et al. [70] emphasize the need to link intervention content to broader contextual determinants of health behaviour. This scoping review found limited explicit integration of environmental barriers or facilitators into game design, a notable omission considering that environmental restructuring is a key function of many successful digital health interventions [71].

### 4.3. Motivation

Motivation is arguably the most multifaceted of the COM-B constructs and encompasses both reflective motivation (for example, intentions, goals) and automatic motivation (for example, emotional responses, habits). In this review, reflective motivation was often activated through improved knowledge, attitudes, and perceived seriousness of cancer risk. For example, public games increased awareness of vulnerability and severity [41], while patient-facing games improved treatment adherence and health decision-making [31]. Among healthcare professionals, games improved confidence and intention to use knowledge in clinical practice [54,58]. Automatic motivation was less directly measured but was suggested through high levels of game satisfaction, enjoyment, and engagement [33,50]. Emotional engagement was a facilitator in many studies, but distress was also noted. For instance, cancer patients expressed concerns that some game content was emotionally overwhelming prior to clinical appointments [38]. This highlights the tension between engaging emotional systems for motivation and the risk of emotional overload. As Albarracín et al. [72] note, affective triggers can be potent motivators, but if misaligned with user readiness, they may inhibit rather than promote behavioural change. It is also noteworthy that engagement, as a proxy for motivation, was consistently high across patient and public games, though some attrition was observed in younger cancer patients, potentially due to treatment-related fatigue [33]. These patterns mirror findings from behavioural science literature showing that motivation must be continually reinforced through reward, feedback, and progress cues [68].

### 4.4. Interactions Across COM-B Constructs

The COM-B model posits that the three components are interdependent. The review provides several illustrations of this interaction. In patient games, improved knowledge (capability) led to higher self-efficacy (motivation), which in turn influenced adherence (behaviour). In public games, increased enjoyment (automatic motivation) facilitated higher learning retention (capability), which improved future intentions to screen (reflective motivation). However, where capability was enhanced but opportunity was constrained, such as lack of time or technological access, behavioural translation was not always assured. Furthermore, despite evidence of strong capability and motivation, few studies demonstrated sustained behaviour change, suggesting that interventions may lack sufficient repetition, follow-up, or integration into broader health systems. This aligns with recent research [66,73], which caution that unless digital interventions are embedded within supportive ecosystems, their behavioural impact may be limited or short-lived.

### 4.5. Implications for Design and Implementation

The results support the utility of the COM-B model for both understanding and designing serious game interventions. However, few studies explicitly referenced behavioural theory in their design. Future development could apply COM-B and associated frameworks (for example, the Behaviour Change Wheel) to systematically identify behavioural targets and intervention functions [62,63,64,65,66]. Moreover, game features should be aligned with behaviour change techniques, as mapped in the Behaviour Change Technique Ontology [62,63,64] to enhance transparency, replicability, and precision in intervention design. From an implementation science perspective, successful deployment will require integration with healthcare delivery systems, especially for patient and professional interventions. As Damschroder et al. [69] argue, the effectiveness of digital tools is not only determined by their content but also by how well they are adopted, implemented, and sustained in practice. In summary, the thematic findings of this review, when interpreted through the COM-B lens, offer valuable insights into the behavioural mechanisms underpinning serious game interventions for cancer awareness and education. While most interventions demonstrate promise in enhancing capability and motivation, fewer address opportunity or systematically integrate behavioural science principles. As digital health tools continue to expand, greater attention to theoretical grounding, social context, and sustained engagement will be essential to ensure their impact on cancer-related behaviours.

Overall, this review shows that digital serious games predominantly act by improving users’ capability and motivation, with more limited attention to the social and environmental opportunities required to sustain behaviour change. Emotional engagement and interactivity are central to their effectiveness but must be carefully balanced to avoid cognitive or emotional overload in sensitive cancer contexts. Future research should more explicitly apply behavioural theory during intervention design, adopt user-centred and culturally responsive approaches, and assess long-term outcomes within real-world healthcare systems. Strengthening these areas will support the translation of serious games into credible, sustainable tools for cancer education and behavioural support.

### 4.6. Strengths and Limitations

This scoping review was conducted in accordance with the Joanna Briggs Institute (JBI) methodology and reported using the PRISMA-ScR checklist to ensure methodological rigour, transparency, and reproducibility [25,30]. The inclusion of a comprehensive search strategy across four major databases (MEDLINE, PsycINFO, Web of Science, and CINAHL) facilitated a broad evidence base covering diverse study designs, including qualitative, quantitative, mixed-methods, co-design, and design-based research. This methodological breadth enabled the capture of complex and varied insights into the use of digital serious games for cancer awareness and education. A key strength of this review lies in its structured approach to analysis and synthesis. The findings were thematically organised around three key target populations: individuals with cancer, healthcare professionals, and members of the public. This structure allowed for meaningful comparison of outcomes and experiences across these stakeholder groups, which are often examined in isolation in the existing literature. The integrative use of the COM-B model as a theoretical framework further strengthened the interpretive depth of the review, enabling a nuanced discussion of how digital serious games might influence behaviour through enhanced capability, opportunity, and motivation [63].

However, there are limitations that should be acknowledged. First, consistent with scoping review methodology, no formal appraisal of study quality was undertaken. While not required for scoping reviews [30], the absence of critical appraisal limits the ability to make judgments about the strength of the evidence and may obscure the relative reliability of findings. Future reviews or meta-analyses that incorporate formal quality assessments could help determine the robustness and transferability of specific intervention outcomes. Second, while the COM-B model provided a valuable lens for interpreting behaviour change mechanisms, applying a theoretical framework retrospectively may limit the consistency of alignment between primary data and the model’s constructs [62,63,64,65,66]. Furthermore, the heterogeneous nature of the included populations posed challenges for direct comparisons. Some studies involved mixed samples, making it difficult to isolate outcomes by group, while others provided minimal demographic or contextual detail. These inconsistencies may have affected the ability to draw group-specific conclusions or identify differential impacts of interventions across age, gender, socioeconomic, or cultural lines. Additionally, the scope was limited to studies published in English, which may have led to the omission of relevant findings from non-English-speaking regions. The exclusion of grey literature was also a limitation, which may have introduced publication and language bias. Therefore, some relevant evidence may not have been captured. To partially mitigate this limitation, backward and forward citation tracking was conducted for all included studies to identify additional eligible records. Future reviews could address these limitations by including grey literature and non-English sources to capture a broader and more diverse range of evidence.

The review also focused exclusively on digital serious games, thereby excluding other game-based or non-digital interventions that may be used for similar educational or awareness-raising purposes. While this focus was intentional and aligns with the increasing digitisation of health education, it may have excluded valuable empirical data from analogue or hybrid game approaches that are more accessible in low-resource settings or among digitally excluded populations [74,75]. Although the review was limited in scope, we attempted to address these issues by systematically grouping studies by serious game type, cancer type, and population, and by indicating any variation in design or outcomes across different contexts. This approach was intended to provide a more nuanced understanding of the evidence. Another limitation concerns the breadth of the concept of “cancer awareness”. While inclusive, the broad categorisation across cancer types, stages, and outcomes may have masked condition-specific nuances. For example, interventions targeting early detection of skin cancer may require fundamentally different design principles than those supporting self-management in advanced-stage cancer patients. The thematic synthesis attempted to account for this variation, but further research focusing on condition-specific serious games is warranted. Finally, the review was limited by a predominance of studies from high-income countries, which may limit generalisability to low- and middle-income settings. Given the increasing global burden of cancer and digital health inequities, there is a need to examine how serious games function across different cultural, economic, and healthcare contexts.

Despite these limitations, this review offers a valuable synthesis of current evidence on digital serious games in cancer awareness and provides a foundation for future research. Integrating theoretical perspectives such as the COM-B model not only strengthens the conceptual rigour of the findings but also helps to guide intervention development and implementation in real-world settings. Future work should prioritize rigorous evaluation, longer-term follow-up, greater diversity in study populations, and explicit use of theory-driven frameworks to inform design and evaluation.

## 5. Conclusions

This scoping review highlights the emerging potential of digital serious games as effective interventions for cancer awareness, education, and behaviour change across varied populations. The evidence suggests that these games were associated with improvements in knowledge, enhance user engagement, support communication between patients and professionals, and promote preventive health behaviours. Positive outcomes were reported among people living with cancer, healthcare professionals, and members of the public, with serious games generally perceived as acceptable, usable, and engaging. These findings support the growing role of serious games in the digital health landscape, particularly in the context of cancer prevention and self-management.

Despite these encouraging findings, important limitations were identified. Evidence was largely drawn from studies with heterogeneous designs and predominantly short-term outcomes. Issues such as user engagement, personalisation, digital accessibility, and sustained impact remain insufficiently addressed. Furthermore, there is limited insight into the effectiveness of specific game components or mechanisms. By applying the COM-B model, this review provided a structured understanding of how serious games may influence behaviour through capability, opportunity, and motivation. Future research should build on this theoretical foundation, integrating robust evaluation methods, user-centred design, and implementation-focused approaches to enhance the reach, effectiveness, and long-term value of digital serious games in cancer education and awareness.

## Figures and Tables

**Figure 1 cancers-17-03368-f001:**
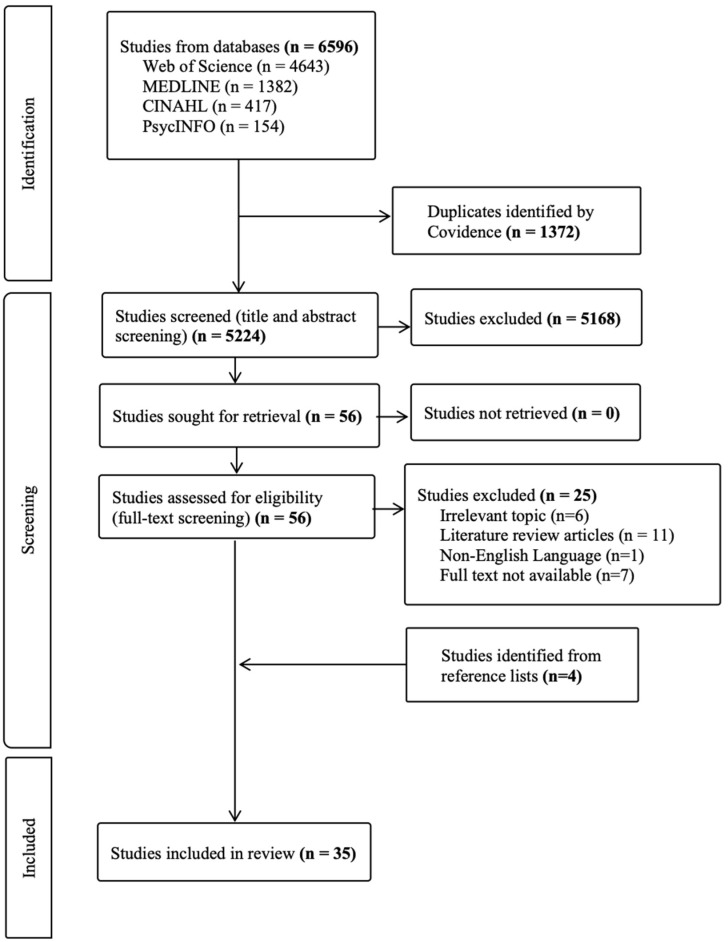
PRISMA flow diagram.

**Figure 2 cancers-17-03368-f002:**
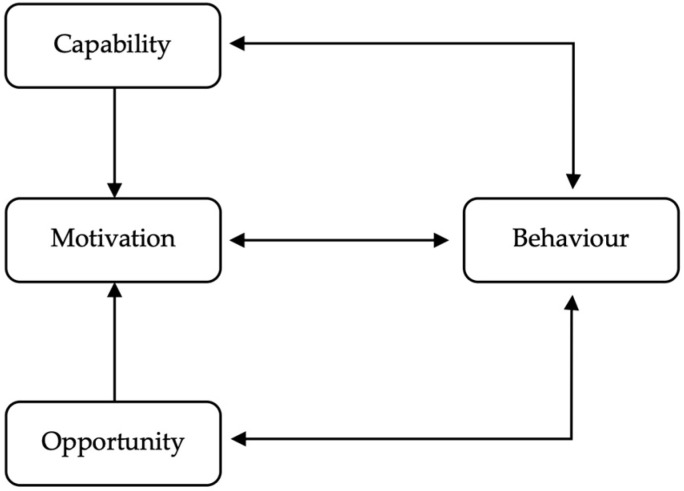
COM-B Model.

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
