# Peer review of "Digital Serious Games for Cancer Education and Behavioural Change: A Scoping Review of Evidence Across Patients, Professionals, and the Public"

_cancers, 2025, doi:10.3390/cancers17203368_

Round 1

Reviewer 1 Report

Comments and Suggestions for Authors

Dear Authors,

Thank you for the opportunity to review your manuscript. I consider that addresses a timely question, synthesises a dispersed literature across patients, professionals, and the public, and commendably frames findings using the COM-B model. I believe the paper is promising and could make a useful contribution after substantive revisions that improve methodological transparency, internal consistency, and the precision of claims.

First, he abstract reports searches of MEDLINE, Embase, CINAHL, PsycINFO, and Scopus, whereas the Methods describe MEDLINE, PsycINFO, Web of Science, and CINAHL, with the formal strategy adapted for MEDLINE and the systematic run dated August 10, 2025. These lists should match exactly across Abstract, Methods, and Supplementary Materials, and the final list should be reflected in the PRISMA diagram and counts. At present, the discrepancy could confuse readers and complicate reproducibility. Consider adding the full, database-specific strategies to the Supplementary Materials (as you have done for MEDLINE) and confirming the de-duplication workflow and yield at each step.

Second, strengthen reporting to fully meet JBI and PRISMA-ScR expectations. You note prospective protocol registration on OSF and adherence to PRISMA-ScR; however, please provide or link the completed PRISMA-ScR checklist, ensure the PRISMA flow diagram is rendered (it is referenced but not visible in this version), and include the precise date ranges covered and any alerting/updates before submission. In addition, please state the number of reviewers at each screening stage and how conflicts were resolved, and provide the reasons for full-text exclusions in an appendix.

Third, be explicit and consistent about eligibility and the PCC framework. The manuscript limits the population to adults and excludes studies in children and adolescents, yet several included trials involve adolescents and young adults (e.g. 13-29 years). Clarify how studies with mixed age bands were handled and ensure all included items meet your stated adult-only criterion or that you justify exceptions (e.g. ≥50% adults with extractable adult data). A short paragraph in Methods explaining this rule, aligned with your PCC decisions, will preempt reader confusion.

Fourth, temper causal language and qualify “effectiveness” claims consistently. Many sections appropriately describe heterogeneity and short follow-up, but phrasing such as “Serious games were effective for improving health and behavioural outcomes” appears as a general conclusion drawn from mixed designs (RCTs, quasi-experimental, qualitative). Where you summarise benefits derived from randomised trials (e.g. adherence gains in adolescents and young adults), you can retain “improved” with a citation; elsewhere, prefer formulations like “were associated with improvements” or “participants reported increases,” and, where possible, provide the effect size ranges or directionality to anchor the narrative. This applies across patient, professional, and public sections and again in the Conclusions.

Fifth, make the COM-B mapping more concrete. You frame the synthesis through COM-B in the Abstract and Discussion; readers would benefit from a compact table that aligns specific game features and outcomes from included studies with Capability, Opportunity, and Motivation, indicating the study design and population for each cell. This will convert a largely narrative mapping into something readers can act on and will also help illuminate where “Opportunity” is under-addressed in the current evidence base.

Sixth, expand the description of study overlap and data aggregation. You note that 35 articles represent 33 unique studies due to shared datasets; please make explicit how you avoided double-counting in narrative summaries and any quantitative tallies (e.g., when stating how many studies support a given theme). A short subsection under Results or Data Charting, clarifying your deduplication at the study level, would suffice.

Seventh, enhance transparency around exclusions and potential bias. You consciously exclude grey literature and non-English publications. Please discuss how these choices might bias the corpus, particularly for serious games that are often disseminated in conference proceedings, theses, or non-indexed repositories, and whether you conducted backwards/forward citation chasing to partially mitigate this limitation. A brief note on publication bias and selective reporting in this fast-moving digital space would also strengthen the Limitations.

Eighth, ensure figures, tables, and supplements are complete and cross-referenced. The PRISMA figure and COM-B illustration are referenced but not rendered in the current PDF; confirm they will appear in the production files. In the Supplementary Materials, the MEDLINE strategy is helpful; parallel strategies (or at least exact query blocks) for each database listed in Methods should be added for reproducibility. Also consider moving very dense descriptive material from the main text to Supplementary Materials 2 and tightening corresponding in-text summaries.

Ninth, refine the Results to balance breadth with parsimony. The present structure is clear and stakeholder-oriented, but some subsections read as extended narrative vignettes. Where feasible, foreground the quantitative backbone, study counts by design, typical sample sizes, key outcomes with direction and, when available, magnitude, and relegate illustrative examples to shorter, citated sentences. This will make the thematic synthesis feel more systematic and less anecdotal.

Tenth, attend to reference accuracy and web citations. For example, the WHO fact sheet includes an access date; ensure the journal’s style for URLs and access dates is followed consistently, and verify items currently labelled as “preprint” that are now published. A light reference audit will prevent production delays.

Author Response

Thank you for your valuable suggestions. I have made some changes accordingly. The revisions that I made were showed as red font. 

First, the abstract reports searches of MEDLINE, Embase, CINAHL, PsycINFO, and Scopus, whereas the Methods describe MEDLINE, PsycINFO, Web of Science, and CINAHL, with the formal strategy adapted for MEDLINE and the systematic run dated August 10, 2025. These lists should match exactly across Abstract, Methods, and Supplementary Materials, and the final list should be reflected in the PRISMA diagram and counts. At present, the discrepancy could confuse readers and complicate reproducibility. Consider adding the full, database-specific strategies to the Supplementary Materials (as you have done for MEDLINE) and confirming the de-duplication workflow and yield at each step.

  1. The databases listed in the Abstract, Methods, Supplementary Material 2, and the PRISMA flow diagram have been checked and corrected to ensure consistency. The four databases were Web of Science, Medline, CINAHL, and PsycINFO.
  2. The complete search strategy for each database (Web of Science, Medline, CINAHL, and PsycINFO) were added in the supplementary material 2.
  3. The de-duplication workflow and yield at each step were illustrated in the 4. Selection of Sources of Evidence.

Second, strengthen reporting to fully meet JBI and PRISMA-ScR expectations. You note prospective protocol registration on OSF and adherence to PRISMA-ScR; however, please provide or link the completed PRISMA-ScR checklist, ensure the PRISMA flow diagram is rendered (it is referenced but not visible in this version), and include the precise date ranges covered and any alerting/updates before submission. In addition, please state the number of reviewers at each screening stage and how conflicts were resolved, and provide the reasons for full-text exclusions in an appendix.

  1. The completed PRISMA-ScR checklist have been provided in the Supplementary Material 1.
  2. The PRISMA flow diagram (Figure S1.) were illustrated in the Results section and have also been added in the Supplementary Materials.
  3. The precise date ranges covered and any alerting/updates before submission have been added in 3. Search strategy.
  4. The number of reviewers at each screening stage and how conflicts were resolved were included in 4. Selection of sources of evidence. Appendix A have been added to provide the reasons for full text exclusions in 3. Results.

Third, be explicit and consistent about eligibility and the PCC framework. The manuscript limits the population to adults and excludes studies in children and adolescents, yet several included trials involve adolescents and young adults (e.g. 13-29 years). Clarify how studies with mixed age bands were handled and ensure all included items meet your stated adult-only criterion or that you justify exceptions (e.g. ≥50% adults with extractable adult data). A short paragraph in Methods explaining this rule, aligned with your PCC decisions, will preempt reader confusion.

  1. A statement of explaining the population eligibility criteria was added to 2.1.1. Population.

Fourth, temper causal language and qualify “effectiveness” claims consistently. Many sections appropriately describe heterogeneity and short follow-up, but phrasing such as “Serious games were effective for improving health and behavioural outcomes” appears as a general conclusion drawn from mixed designs (RCTs, quasi-experimental, qualitative). Where you summarise benefits derived from randomised trials (e.g. adherence gains in adolescents and young adults), you can retain “improved” with a citation; elsewhere, prefer formulations like “were associated with improvements” or “participants reported increases,” and, where possible, provide the effect size ranges or directionality to anchor the narrative. This applies across patient, professional, and public sections and again in the Conclusions.

  1. We revised several conclusive sentences throughout the manuscript to use neutral expressions.

Fifth, make the COM-B mapping more concrete. You frame the synthesis through COM-B in the Abstract and Discussion; readers would benefit from a compact table that aligns specific game features and outcomes from included studies with Capability, Opportunity, and Motivation, indicating the study design and population for each cell. This will convert a largely narrative mapping into something readers can act on and will also help illuminate where “Opportunity” is under-addressed in the current evidence base.

  1. Each included study was mapped to the COM-B model. The detail of the COM-B components of each study has been added in the data characteristics table (supplementary material 3). An introduction was added in 4. Discussion.

Sixth, expand the description of study overlap and data aggregation. You note that 35 articles represent 33 unique studies due to shared datasets; please make explicit how you avoided double-counting in narrative summaries and any quantitative tallies (e.g., when stating how many studies support a given theme). A short subsection under Results or Data Charting, clarifying your deduplication at the study level, would suffice.

  1. The description of how to avoid double-counting has been added in 5. Data charting and 3. Results.

Seventh, enhance transparency around exclusions and potential bias. You consciously exclude grey literature and non-English publications. Please discuss how these choices might bias the corpus, particularly for serious games that are often disseminated in conference proceedings, theses, or non-indexed repositories, and whether you conducted backwards/forward citation chasing to partially mitigate this limitation. A brief note on publication bias and selective reporting in this fast-moving digital space would also strengthen the Limitations.

  1. We have added a paragraph in the 6. Strengths and limitations discussing this limitation and approach of addressing the potential bias.

Eighth, ensure figures, tables, and supplements are complete and cross-referenced. The PRISMA figure and COM-B illustration are referenced but not rendered in the current PDF; confirm they will appear in the production files. In the Supplementary Materials, the MEDLINE strategy is helpful; parallel strategies (or at least exact query blocks) for each database listed in Methods should be added for reproducibility. Also consider moving very dense descriptive material from the main text to Supplementary Materials 2 and tightening corresponding in-text summaries.

  1. The PRISMA flow diagram (Figure S1.) and COM-B Model (Figure S2.) were both added to the supplementary materials.
  2. The complete search strategy for all four databases (Supplementary material 2) were added in the supplementary materials.
  3. We appreciate the suggestion. Some of the descriptive data were retained in the main text because they provide important contextual support for the interpretation of findings and make the conclusions clearer and more accessible to readers.

Ninth, refine the Results to balance breadth with parsimony. The present structure is clear and stakeholder-oriented, but some subsections read as extended narrative vignettes. Where feasible, foreground the quantitative backbone, study counts by design, typical sample sizes, key outcomes with direction and, when available, magnitude, and relegate illustrative examples to shorter, citated sentences. This will make the thematic synthesis feel more systematic and less anecdotal.

  1. The quantitative summarise and shortened narrative descriptions were added in Results, for example, the first paragraph in 3.2.1, 3.2.2.

Tenth, attend to reference accuracy and web citations. For example, the WHO fact sheet includes an access date; ensure the journal’s style for URLs and access dates is followed consistently, and verify items currently labelled as “preprint” that are now published. A light reference audit will prevent production delays.

  1. The reference list has been checked and revised according to the requirement of the journal.

Reviewer 2 Report

Comments and Suggestions for Authors

Many thanks to the authors for the work presented. The topic is of interest and has the potential to contribute to the exiting body of research. 

Please find the attached file with some comments which may help increase the value to the readers an the impact of the review.

A stronger critical approach e.g., adding small conclusions box (take home messages) at the end of each chapters, some visuals to correlate the concepts presented and the data, substantial references to highlight the critical discourse in the field could help. 

Author Response

Thank you very much for taking the time to provide your thoughtful comments and suggestions. The revisions that I made were showed as red font. 

Please find the attached file with some comments which may help increase the value to the readers an the impact of the review.

We have carefully considered the comments in the attached file and have made revisions accordingly.

A stronger critical approach e.g., adding small conclusions box (take home messages) at the end of each chapters, some visuals to correlate the concepts presented and the data, substantial references to highlight the critical discourse in the field could help. 

A Highlight section at the end of the 1. Introduction was added to provide some key points for the readers.

Reviewer 3 Report

Comments and Suggestions for Authors

This study reviewed the map evidence on serious games for cancer prevention, care, and survivorship among the public, patients, and healthcare professionals, framed through
the COM-B model. The comments are as follows:

  1. In 2.3 searching strategy, have the authors applied the similiar of different approach in searching PsycINFO, Web of Science, and CINAHL, comparing to searching MedLine?
  2. In 2.6 data analyses, have the authors used specific statistical software package? If yes, the name and version of the package should be noted. 
  3. Line 286: , “The Republic of China (Taiwan)" should be revised as: "Taiwan, Province of China".
  4. Line 644: "Figure 2. COM-B Model", please relocate them into the results part.
  5. Abbreviations (line 834-849) should be listed following A-Z order.
  6. The format of references need an in-depth revision according to the requirement of the journal. E.g., keeping DOIs or remove them, using full format or abbreviation of names of journal, correctly using capital letters. 

Author Response

Thank you very much for the valuable suggestions. I have made several changes accordingly. The revisions that I made were showed as red font. 

This study reviewed the map evidence on serious games for cancer prevention, care, and survivorship among the public, patients, and healthcare professionals, framed through
the COM-B model. The comments are as follows:

  1. In 2.3 searching strategy, have the authors applied the similiar of different approach in searching PsycINFO, Web of Science, and CINAHL, comparing to searching MedLine?

The complete search strategy for each database (Web of Science, Medline, CINAHL, and PsycINFO) were added in the supplementary material 2.

2. In 2.6 data analyses, have the authors used specific statistical software package? If yes, the name and version of the package should be noted. 

Statistical software package was not used in the data analysis. The data were analysed using a narrative synthesis approach with three sequential phases as listed in 2.6. Data analysis.

3. Line 286: , “The Republic of China (Taiwan)" should be revised as: "Taiwan, Province of China".

This has been revised as ‘Taiwan, province of China’.

4. Line 644: "Figure 2. COM-B Model", please relocate them into the results part.

The Figure S2. COM-B Model was added in 4. Discussion and also in the supplementary material.

5. Abbreviations (line 834-849) should be listed following A-Z order.

Abbreviations have been reordered following A-Z.

6. The format of references need an in-depth revision according to the requirement of the journal. E.g., keeping DOIs or remove them, using full format or abbreviation of names of journal, correctly using capital letters. 

      The reference list has been checked and revised according to the requirement of the journal.

Round 2

Reviewer 1 Report

Comments and Suggestions for Authors

Dear authors

Thanks for submitting the revised version of your manuscript. The changes included improving the scientific quality of your manuscript 

Author Response

Thank you for this supportive feedback

Reviewer 2 Report

Comments and Suggestions for Authors

Many thanks to the authors for their work on the manuscript. 

Since it is a review, and the audience may wish to more easily retain the critical information (take-home messages) that the authors considered essential about the topic presented, may the authors add the specific learning points for each chapter at the end of chapters 2, 3, and 4?  

Thank you and all the best.

Author Response

We agree that readers benefit from clear summaries of the key messages in each section. To address this, we have refined the concluding paragraphs of Chapters 2, 3, and 4 to more explicitly summarise the essential findings and implications. These changes have been made in blue font within the manuscript:

  • Collectively, these methods ensured that the evidence was comprehensively mapped and that analytical transparency was maintained throughout the review process. The approach provided a structured foundation for identifying key patterns across study designs and populations, ensuring that the synthesis accurately reflected the scope of current research on digital serious games for cancer education and behavioural change.
  • Taken together, the results demonstrate that digital serious games have been implemented across a wide range of cancer types, populations, and platforms, with generally positive outcomes for knowledge, engagement, and self-efficacy. Evidence indicates that these interventions can enhance patient understanding and adherence, strengthen professional education and communication, and improve public awareness and preventive intentions. However, the overall evidence base remains heterogeneous, and few studies have examined sustained behavioural change or long-term effects. The findings therefore suggest considerable potential for digital serious games as an emerging strategy for cancer education, while highlighting the need for more rigorous evaluation and theoretical integration.
  • Overall, this review shows that digital serious games predominantly act by improving users’ capability and motivation, with more limited attention to the social and environmental opportunities required to sustain behaviour change. Emotional engagement and interactivity are central to their effectiveness but must be carefully balanced to avoid cognitive or emotional overload in sensitive cancer contexts. Future research should more explicitly apply behavioural theory during intervention design, adopt user-centred and culturally responsive approaches, and assess long-term outcomes within real-world healthcare systems. Strengthening these areas will support the translation of serious games into credible, sustainable tools for cancer education and behavioural support.